# A successional shift enhances stability in ant symbiont communities
**Thomas Parmentier** [1,2] ✉, **Dries Bonte** [1] & **Frederik De Laender** [2]

Throughout succession, communities undergo structural shifts, which can alter the relative abundances of species and how they interact. It is frequently asserted that these alterations beget stability, i.e. that succession selects for communities better able to resist perturbations. Yet, whether and how alterations of network structure affect stability during succession in complex communities is rarely studied in natural ecosystems. Here, we explore how network attributes influence stability of different successional stages of a natural network: symbiotic arthropod communities forming food webs inside red wood ant nests. We determined the abundance of 16 functional groups within the symbiont community across 51 host nests in the beginning and end stages of succession. Nest age was the main driver of the compositional shifts: symbiont communities in old nests contained more even species abundance distributions and a greater proportion of specialists. Based on the abundance data, we reconstructed interaction matrices and food webs of the symbiont community for each nest. We showed that the enhanced community evenness in old nests leads to an augmented food web stability in all but the largest symbiont communities. Overall, this study demonstrates that succession begets stability in a natural ecological network by making the community more even.

Ecological networks represent the various ways in which species interact. Natural communities often host a rich diversity of interaction types[1] and predicting community dynamics, composition, and stability requires knowledge of these interactions[1–5]. Stability is a multidimensional concept[6], and several theoretical and empirical studies have identified various factors that influence different aspects of stability in ecological networks, all linked to their specific structure such as species richness, connectance, as well as diversity, and the kind of species interactions[7–9]. Other aspects relate to how abundances are distributed across species. Specifically, higher evenness may enhance different kinds of stability, such as the maintenance of ecosystem function[10] or resistance to invasion[7].

Succession, described as the process of repeated assembly over time[11] is of fundamental significance in community ecology[12]. Succession typically focuses on how diversity and community composition change, and empirical studies have shown that habitat specialization often differs between early successional and late successional species. Intuitively, and as confirmed for bird[13] and plant communities[14], higher specialization is expected at later stages (but see ref. 15). In general, succession involves more than changes in community composition and specialization and can profoundly shift species interactions and cause rewiring of the ecological network (symbiont web[16–18], food web[19–22]).

Changes in network structural attributes can play a role in preserving network stability across a succession gradient[23], but the precise mechanisms through which successional changes in network structure enhance community stability are to date poorly understood. Available studies suggest that communities become more stable as succession progresses. While intuitively appealing, this conclusion is often based on observations of network structural attributes that may affect stability (e.g., increasing diversity, nestedness, modularity)[18,24] rather than on stability itself[25]. Importantly, these studies also do not account for evenness, which if it changes during succession may influence stability, too. Given the relevance of disturbances in both setting back successional stages or changing local species abundances from the different stages of community assembly, a profound understanding of evenness and its effect on stability is needed.

In this study, we characterize the successional changes in species abundances and network interactions in a complex community of arthropod symbionts living in the nests of red wood ants and examine how these changes affect stability. We define stability using arguably the most frequently-used approach: through analysis of the community matrix's eigenvalues, which measures to what extent a community can recover from perturbations of abundance[2,26]. Our results indicate that there is a significant reorganization of the symbiont community between new and old nests,

[1]Terrestrial Ecology Unit, Department of Biology, University of Ghent, Ghent, Belgium. [2]Research Unit of Environmental and Evolutionary Biology, naXys, ILEE, University of Namur, Namur, Belgium. ✉e-mail: thomas.parmentier@ugent.be

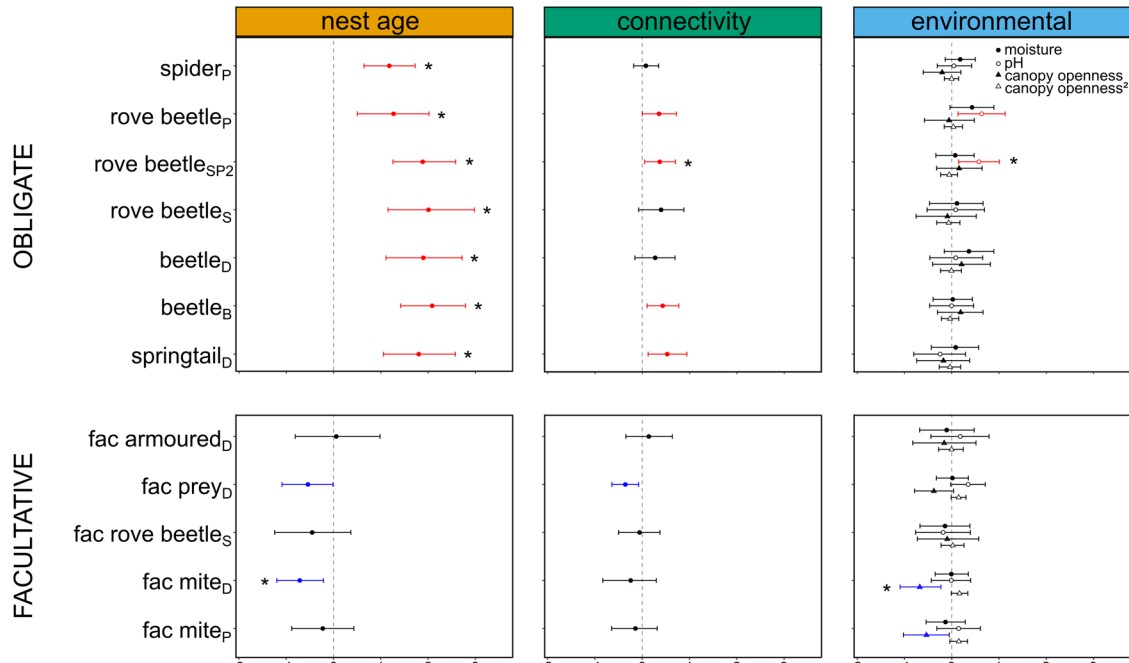

**Fig. 1 | Posterior parameters ($\beta$'s) and 95% credible intervals estimating the responses of the functional groups to the nest predictors (joint species distribution model with lowest WAIC).** Functional groups are categorized into obligate and facultative groups. Nest predictors are categorized into three panels: nest age and connectivity are displayed in a separate panel, whereas the predictor's moisture, pH, canopy openness, and its quadratic effect (denoted by canopy openness[2]) are grouped in the panel "environmental". If the credible interval only contains positive values (colored red), the predictor has a positive effect on the abundance of the functional group with this level of statistical support, if the credible interval only contains negative values (colored blue), the abundance of the functional groups decreases with the predictor. Estimates with 99% posterior probability are indicated with an *.

so succession is indeed the major driver of spatial turnover of composition and network structure in this system. The symbiont community and trophic network associated with new and small nests are dominated by functional groups that are facultatively associated with the ants. Old and large nests contain more even communities with a higher share of obligatorily and more specialized associates. The higher functional group evenness played a crucial role in enhancing the dynamic stability of the symbiont food web associated with the older nests.

## Results

We collected 8367 obligate symbionts and 20,664 facultative symbionts in the 51 red wood ant nests in total.

### Shift in symbiont communities in old nests

A joint species distribution model with nest age, connectivity, moisture, and pH as linear terms and canopy openness as quadratic term had the best fit (model 7: lowest WAIC, Supplementary Table S1, diagnostics Supplementary Information) and was retained for the analyses. The mean explanatory power of the model was $R^2 = 0.55$. The type of association (facultative vs. obligate) of functional groups was strongly correlated with their response to nest age (95% credible interval of $\gamma$ parameter is positive), but not to other covariates (95% credible intervals of $\gamma$ parameters include zero). Indeed, all functional groups obligatorily associated with ants reached higher densities in old nests than in new nests (95% credible interval only contains values greater than zero, Fig. 1). In contrast, no (95% credible interval includes zero) or a negative association (95% credible interval only contains values lower than zero) between nest age and abundance was found in the five facultatively associated groups (Fig. 1). Four obligate groups responded positively to higher connectivity of the nests. Eleven response estimates were supported with at least 99% posterior probability (indicated with * in Fig. 1) including the positive responses of the 7 obligate functional groups to nest age. These results show that nest age is the main predictor of the turnover of the symbiont composition. After controlling for the measured environmental variables and spatial autocorrelation, most obligate functional groups showed positive covariance (Supplementary Fig. S1). This is likely due to similar habitat requirements and/or unmeasured predictors to which the groups show a positive response, rather than by biotic interactions among these functional groups. For example, the abundance of all obligate groups might linearly increase with nest age. Nevertheless, we encountered a limitation in our modeling approach, as we were constrained to categorize nest age into just two classes, rather than treating it as a continuous predictor.

Variance partitioning over the fixed and random effects showed that most explained variation in the abundance of the obligatorily associated groups was attributed to nest age (obligate: 43.6%, facultative: 12.2%), while environmental variables (pH, moisture and canopy openness) were more important in structuring the facultatively associated groups (obligate: 20.0%, facultative 49.4% of the explained variation) (Fig. 2). The remaining variation in both groups was caused by spatial random effects (obligate: 27.0%, facultative: 23.5%) and by connectivity (obligate: 9.6%, facultative: 14.9%) to a lesser degree. Spatial random effects can be attributed to dispersal limitation, co-occurrence, or unmeasured spatially structured variables.

### Communities in old nests are different and more even

The composition of functional groups was significantly different between old and new nests (NMDS plot Supplementary Fig. S2, PERMANOVA, $R^2 = 0.10$, $F = 5.53$, $P = 0.001$, Fig. 3). Old and new nests showed homogeneity of variances (PERMDISP, $F = 1.85$, $P = 0.18$). Community composition of functional groups in red wood ant nests was not affected by connectivity ($R^2 = 0.02$, $F = 1.35$, $P = 0.19$) or its interaction with nest age ($R^2 = 0.02$, $F = 1.16$, $P = 0.29$). Older nests showed a higher evenness $J$ than new nests (mean $J$ old nests = 0.73, 95% CI [0.69–0.76], mean $J$ new nests = 0.55, 95% CI [0.49-0.60], betaregression, $\chi^2 = 31.78$, $P < 0.001$, Supplementary Fig. S3 and Fig. 3). Nest connectivity ($\chi^2 = 0.004$, $P = 0.95$) and the interaction between nest age and nest connectivity ($\chi^2 = 0.75$, $P = 0.39$) did not affect the evenness of symbiont communities. The higher evenness

**Fig. 2 | Variance partitioning of the explained variation among the fixed and random effects of the joint species distribution model.** The functional groups are categorized into obligate and facultative groups.

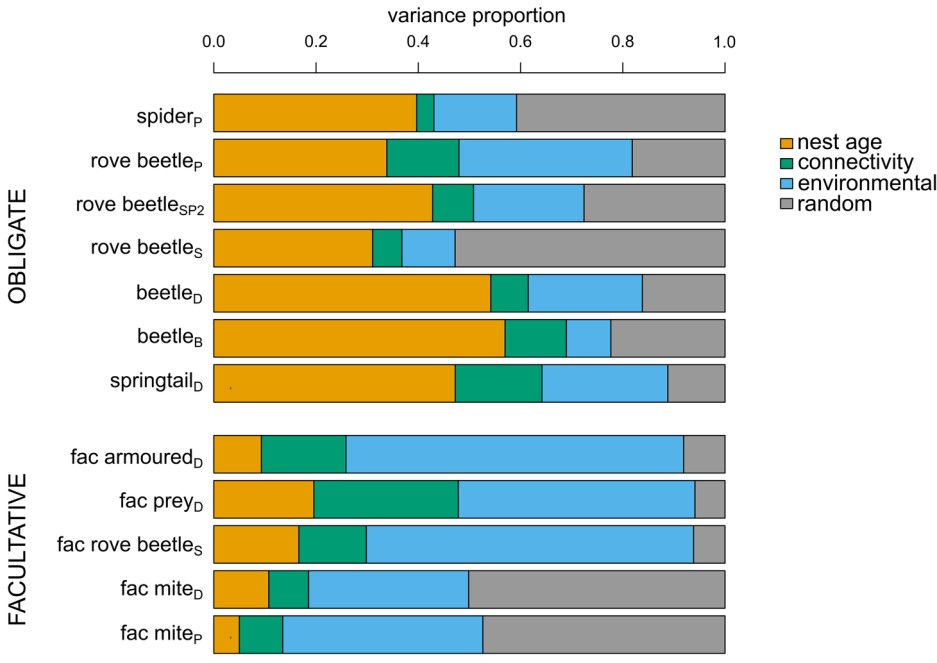

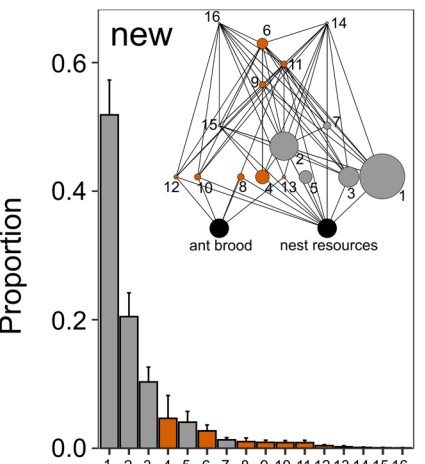
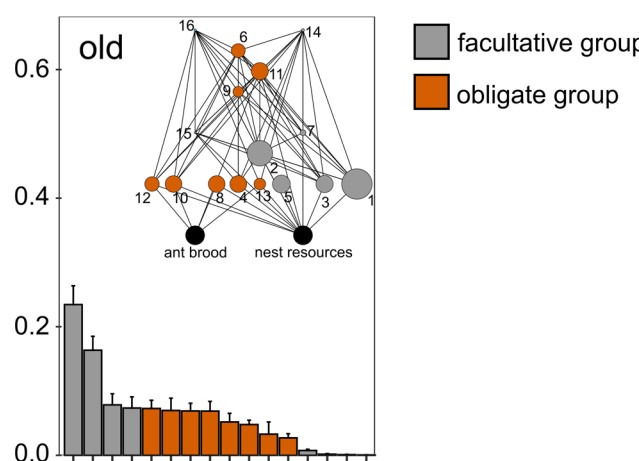

**Fig. 3 | Rank abundance curves and corresponding averaged food webs for symbiont communities in new and old nests.** The rank abundance curves display the relative abundance of the different functional groups. Error bars represent standard errors. The size of the circles in the corresponding food webs is scaled to the average relative abundance of each functional group. Number codes on the x-axis and in the food webs refer to the following functional groups: (1) fac mite$_D$, (2) fac mite$_P$, (3) fac prey$_D$, (4) springtail$_D$, (5) fac armoured$_D$, (6) spider$_P$, (7) fac rove beetle$_S$, (8) beetle$_B$, (9) rove beetle$_P$, (10) beetle$_D$, (11) rove beetle $_{SP2}$, (12) rove beetle$_S$, (13) isopod$_D$, (14) fac pred$_{P1}$, (15) fac pred$_{P2}$, (16) rove beetle$_{SP1}$.

in old nests was mainly caused by the relative higher abundances of the obligate functional groups as displayed in Fig. 3.

## Symbiont food webs in older nests are more stable due to differences in relative abundance

When calculating stability based on the observed relative abundances and the existing topological arrangement of the interaction matrix ($a_{ij} \sim N(\mu,\sigma)$ with $\mu = -0.1$ and $\sigma = 0.05$), we found that new nests were less stable than old nests (GLM, $F = 6.89$, $P = 0.01$, top-left panel Fig. 4). The stability of old and new nests declined with increasing number of functional groups (= community size, GLM, $F = 17.28$, $P < 0.001$), but at a faster rate in old nests (interaction, GLM, $F = 5.31$, $P = 0.03$). Therefore food web stability in old and new nests was not different when they host large symbiont communities. Randomly rewiring the food web (topological arrangement) but keeping the observed relative abundances did not alter the effects of community size and nest age on stability (bottom-left panel Fig. 4). New nests

were less stable (GLM, $P = 0.01$), and both new and old nests showed less stability when accommodating larger communities ($P < 0.001$). By contrast, evening out the abundances of all community members but keeping the topological arrangement of the nodes, we no longer found differences between the stability of old and new nests ($P = 0.72$) (top-right panel Fig. 4). There was a negative effect of community size on stability ($P < 0.001$). If in addition to setting the proportions of the functional groups to evenness, the topology of the interaction matrix was reshuffled, the stability values of old and new nests were also overlapping ($P = 0.74$) and showed a similar negative slope with increasing community size ($P < 0.001$) (bottom-right panel Fig. 4). Taking other $\mu$'s ($a_{ij} \sim N(\mu,\sigma)$) did not change the patterns and results (sensitivity analysis in Supplementary Fig. S4).

## Discussion

Our results show that community structure and the associated food web of red wood ant symbionts are highly different between new and old nests,

**Fig. 4 | Prediction of local stability for symbiont food webs in old and new nests with increasing community size (=number of functional groups) in four scenarios, $a_{ij} \sim N(\mu,\sigma)$ with $\mu = -0.1$ and $\sigma = 0.05$.** The plot in the top-left corner represents the first scenario with the local stability analysis based on the sampled (=observed) relative abundances of the functional groups and the reconstructed (=observed) topological arrangement of the nodes in the food web. In the three other scenarios either the abundance of the functional groups was set even and/or the topological arrangement of the nodes was randomized. Points in the plots indicate local stability: $\log(-\Re(\lambda_1))$, for the 50 permuted community matrices calculated for each nest community. Less negative values indicate higher stability.

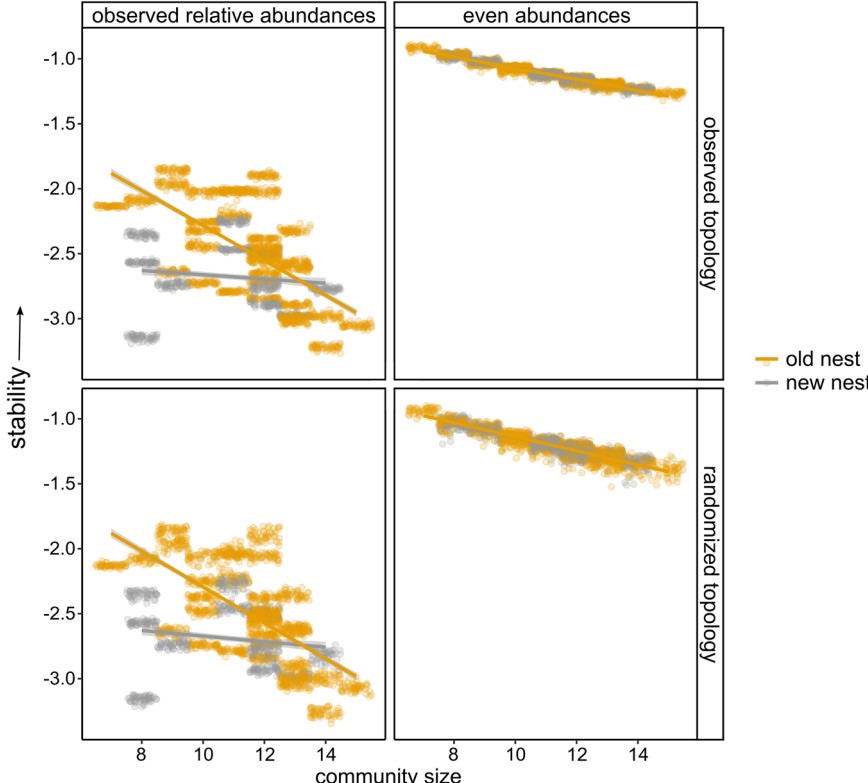

reflecting successional dynamics. There is a strong surge in the number of individuals in all obligate symbiont groups and the community as a whole becomes more even. This shift in relative abundances ultimately leads to higher stability in symbiont networks associated with old host nests, except for large symbiont networks.

The symbiont fauna of red wood ants is known to be influenced by a range of environmental factors. In line with the general predictions of island theory[27], previous work found a higher diversity and/or abundance of symbionts in larger and more connected nests[28–30]. Nest moisture was also found to impact the richness and abundance of different facets of the symbiont fauna in boreal forests[31,32]. Variations in species richness and the abundance of particular groups can offer some insights into how communities change. However, to gain a complete understanding of community succession, it's crucial to shift the focus from calculating species richness alone and instead consider the simultaneous responses of different community components. Some studies could reveal site-specific community differences in groups of symbionts[30,31,33], but there has been no evidence of any influence from environmental factors on the composition of the red wood ant symbiont community (cfr. no land use effect in ref. 32 and no nest size effect in ref. 34). These prior studies, in addition, only focused on the composition of specific groups of arthropod symbionts (such as obligate beetles, mites). By focusing on all groups of symbiotic arthropods within one study site, we here could demonstrate that the symbiont community composition as a whole dramatically shifts as one moves from new, small nests to old, large nests (Figs. 1, 3 and Supplementary Fig. S2).

Successional changes in species composition have been widely studied in different systems including aquatic ponds, islands, or grassland systems and the degree of specialization of the communities may change during succession. So far, varying responses have been found depending on the system, including increased, decreased, or no change in specialization over time[35]. In the red wood ant system, we clearly identified an increase in specialization in older nests (Figs. 1 and 3). Newly founded red wood ant nests are dominated by general soil dwellers that are recruited from the landscape matrix where new nests are founded[36]. While these nests are also quickly colonized by obligate myrmecophiles, their diversity and abundance

are relatively low[37]. In island-like habitats, generalist species have the ability to colonize from adjacent matrix habitats. On the other hand, habitat specialists like obligate myrmecophiles face significantly greater isolation because they cannot establish populations between habitat patches[38]. The community composition as a whole changes when nests age, as the facultative groups do not increase or even decrease in the number of individuals, whereas the number of individuals in all obligate functional groups strongly increases. Ultimately, this different response of obligate and facultative groups leads to an even community with a more dominant role of obligate groups in the food web (Fig. 3). It is important to note that specialization here refers to the degree of association with the host ant, unspecialized species are facultative, specialized species can only live in association with ants. The group of obligate myrmecophiles in itself is very heterogeneous and different levels of behavioral, chemical, and morphological specialization have evolved in a tendency to live in closer intimacy with the ant host[39]. However, most obligate red wood ant symbionts do not exhibit advanced deception strategies[40].

While there is substantial evidence supporting the idea of successional changes in networks, few studies take the additional step of predicting how these changes impact the stability dynamics of networks. We here first demonstrate that in older ant nests, symbiont communities, and their associated food webs (a symbiont community is structured in a localized food web with distinct trophic levels[41,42]) the abundances of all obligate groups increased. Using local stability analyses of the community matrices we then could predict that the composition of the symbiont networks in older nests resulted in higher stability, except for the largest communities. Neutel and colleagues found structural network attributes that preserve stability when networks become more complex along a succession gradient, but could not demonstrate that older networks were more stable for a given level of complexity[23]. Differences in stability between old and new nest networks in our study were entirely attributed to the successional shifts in the relative abundances of distinct functional symbiont groups, eventually resulting in older nest networks becoming more even in abundance. Unlike other studies that found a substantial effect of the arrangement of the interactions within food webs on stability[43,44], changing food web topologies

did not result in significant differences in the stability of old and new nest networks. Consistent with May's predictions[45] and prior theoretical and empirical research[23,46], we also observed an adverse effect of the number of functional groups (community size) or complexity on stability. Interestingly, stability in old nest communities decreased faster with increasing community size than in new nest communities. Consequently, there was no difference in stability between the largest symbiont communities of old and new nests.

Our results uncover how increased evenness and heightened levels of specialization in older natural communities associated with red wood ant nests contribute to their enhanced stability. However, this work also holds significant relevance for conservation efforts. Red wood ant nests are a critical habitat for this unique assemblage of symbionts. Nests are under growing pressure in an era of rapid environmental change and their numbers are declining across Europe[47]. As our understanding of the intricate interplay between these ants and their symbionts deepens, it becomes increasingly evident that safeguarding mature, well-established nests is imperative for the persistence of this unique and hidden network of symbionts.

## Methods
### Study system
The large organic nest mounds of red wood ants are one of the most iconic insect structures found in European forests. A surprisingly diverse, but hidden community of arthropods lives in red wood ant mounds and in the subterranean galleries[48,49]. These ant symbionts, also known as ant inquilines or myrmecophiles, resort to a series of chemical, behavioral, and morphological defense strategies to cope with the hostile ant nest environment[40,50]. Broadly, we can categorize the arthropods living inside red wood ant nests into two categories. Obligate symbionts are species that can only be found in association with ants. Some of them are not specific to red wood ants and can also target other ant species. Apart from a gradient in host specificity, we also find a gradient in specialization in the behavior, chemical ecology, and morphology of this group of obligate symbionts. Facultative symbionts, on the other hand, are common soil organisms that may opportunistically live in the ant nest. They have a loose association with ants and are mostly found away from ants[36]. Although the ant nest environment does not seem their preferred habitat they can become quite abundant[51]. They are less specialized than obligate symbionts and do not possess specific adaptations to live in ant nests. Red wood ant nests form a neat microcosm housing a community of multiple symbiont species that live together and Interac. Many of the symbionts negatively impact their ant host, by feeding on ant broods or by pilfering collected prey. However, the symbiont community is also driven by several trophic interactions among the symbionts[40]. As a result, every red wood ant nest supports a local food web of symbionts encompassing detritivores, scavengers, parasites, and predators. Recent research demonstrated that these local symbiont networks are connected through frequent dispersal[37].

### Study site and sampling
We performed our research in the Sint-Sixtusbossen in Poperinge (N 50.885622, W 2.698785°), north-west of Belgium. This site is a mosaic of different forest fragments intersected by roads and farmland. It holds a polydomous (=multiple mounds/nests) colony of the red wood ant *Formica rufa* Linnaeus, 1761 distributed over 51 nest mounds (in 2020). The nests are mainly situated along the southern edge of the forest fragments. Mounds in the colony cooperate, and exchange food, brood, and workers via trails running between the nests. Every nest mound contains multiple queens (pers. observations TP).

We sampled ant symbionts in all known nests (51 in total) in the study site during July and August 2020. We used rectangular pitfalls (plastic Sunware Q-Line Box: 27×8.4×9 cm, volume: 1.3 L) with a 1 cm layer of moist plaster on the bottom. We positioned three pitfalls deep in every nest mound. The sides of the traps were too slippery for the myrmecophiles to escape from, but ants could easily climb out of these boxes. We covered the

pitfalls with a plastic plate to prevent excessive organic nest material from falling in from above. The plate was positioned 2 cm above the opening of the pitfalls allowing symbionts to enter the pitfall through a wide slit. As ants gradually filled the pitfall with nest material, these boxes had to be emptied every 1–2 days to prevent the symbionts from escaping. The myrmecophiles and nest material were kept apart per nest to avoid double counts. We moistened the plaster if needed and put the empty pitfalls back in the nest but in a different position to maximize sampling of all localities in the nest. After a week, we emptied these boxes. In total, every nest was sampled for 7 days with 3 boxes. For each nest, we spread the organic nest material that was found in the three pitfalls over the week sampling period on a large white tray in the lab. The larger symbionts were directly collected with an aspirator and sorted per species. The organic material was then placed in a Berlese trap for 5 days until the material was completely dry. Small facultative symbionts such as springtails, mites, and juveniles of isopods and diplopods were mainly collected with this method. Obligate symbionts were identified at the species level, facultative symbionts were grouped in different functional groups. In the end, we discriminated 9 obligate functional groups based on their taxonomy and feeding ecology[41,52]. Type of feeding ecology is indicated by subscript: $_P$: predator, $_S$: scavenger, $_{SP}$: scavenging and preying, $_D$: detritivore, $_B$: brood predator): spider$_P$: *Thyreosthenius biovatus* ; rove beetle$_P$: *Stenus aterrimus* ; rove beetle$_{SP1}$: *Quedius brevis* ; rove beetle$_{SP2}$: *Thiasophila angulata* ; rove beetle$_S$: *Amidobia talpa, Dinarda maekelii, Notothecta flavipes* and *Lyprocorrhe anceps* ; beetle$_D$: *Monotoma angusticollis* and *Monotoma conicicollis* ; beetle$_B$: *Clytra quadripunctata* ; springtail$_D$: *Cyphoderus albinus* ; isopod$_D$: *Platyarthrus hoffmannseggii*. Species that are taxonomically related but exhibit distinct trophic ecologies are categorized into separate functional groups: for example, rove beetles are assigned to rove beetle$_P$, rove beetle$_S$, rove beetle$_{SP1}$, or rove beetle$_{SP2}$ based on their specific diet (functional group-specific diet in Supplementary Table S2). The rare obligate species *Myrmetes paykulli* ($N = 4$), *Pella humeralis* ($N = 7$), *Emphylus glaber* ($N = 5$), and *Leptacinus formicetorum* ($N = 2$) were excluded from our analyses to limit the number of obligate functional groups. By doing so, we avoided convergence issues in our analyses and reduced the complexity of the theoretical stability analyses (see below). Next, we grouped the facultative symbionts into 7 functional groups on the basis of their taxonomy and diet (derived from literature[53]): fac armoured$_D$, detritivorous groups with a protective hard exoskeleton: Isopoda (except the obligate *P. hoffmannseggii*) and Diplopoda ; fac pred$_{P1}$: Chilopoda ; fac pred$_{P2}$: Diplura ; fac prey$_D$: Diptera, Collembola (except the obligate *C. albinus*), Psocoptera and Thysanoptera ; fac rove beetle$_S$: Staphylinidae ; fac mite$_D$: Astigmata and Oribatida ; and fac mite$_P$: Uropodina, Gamasina and Prostigmata. Rare facultative groups (=groups with <15 individuals overall nests, e.g., Heteroptera, Hymenoptera, Lepidoptera, these groups compromised a total of 86 individuals) were not included to limit the number of facultative functional groups. In addition, the trophic interactions of these groups were mostly unclear. The trophic ecology and trophic interactions of the different functional groups are given in Supplementary Table S2.

### Predictors
The well-delineated nature and the significant variation in age, connectivity, and environmental conditions of red wood ant nests make them excellent natural and closed systems for studying how different groups of associated organisms respond and how symbiont communities are assembled over time and along environmental gradients[54]. For each of the 51 nests, we measured different parameters that may affect the assembly of the symbiont communities: nest surface, activity, nest age, connectivity, moisture, pH, and canopy openness. We used the nest surface (range: 0.3–7.3 m²) as a proxy for the size of the red wood ant microcosm ('ecosystem size'). Nest activity was defined as the rate of workers entering and leaving the nest. We placed plastic plates (12 × 30 cm) around the entire nest on the soil (ants could not go under), left it for 10 min, and knocked the ants off in a container with the wall coated with an anti-escape layer (fluon). The total biomass of the ants removed from the plates was then used as a proxy for nest activity. These

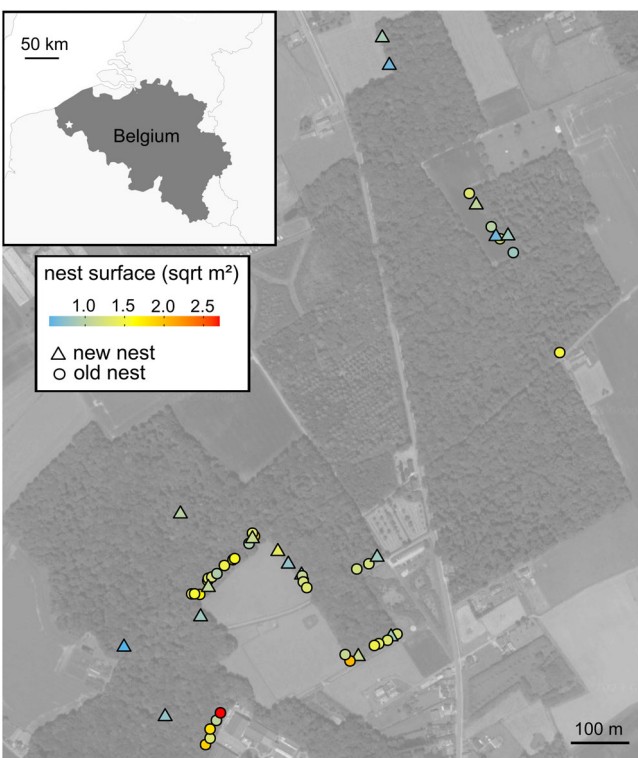

**Fig. 5 | Spatial distribution of the red wood ant nests in the study site in Northwest Belgium.** Nests are classified according to their age: new nests are those established for less than 1.5 years (triangles), while old nests are over 5 years old (circles). The color gradient indicates the square-rooted nest surface as a proxy for nest size. There is a strong positive correlation between nest surface and nest age ($r = 0.61$).

activity measures were all done on two cloudy days between 11 and 15 pm in the last week of August, 2020. As we monitored the nests over the last 10 years[52], we have a clear understanding of the distribution and age of the different nests. We grouped nests into two age classes: new nests that were founded in 2020 or in 2019 (less than 1.5 years old), and old nests that were in the same position for at least 5 years. The connectivity of a nest measured the proximity to other red wood ant nests. The connectivity of a nest was calculated as the number of old red wood ant nests within a 100-m radius of the nest. We measured in the lab on a subsample of the organic thatch material of the nest (ca. 10 g) the pH (1:5 dry organic material:water suspension, pH sensor InLab Mettler Toledo). We also determined the moisture (1-dry sample weight/wet sample weight) of organic nest material taken from the core of the nest. Lastly, we assessed the openness of the vegetation around and above the nests. We placed a tray of which the outer borders were coated with fluon on top of each mound. Then we put a camera (Nikon D5200, 24.1 megapixels, dynamic range of 12.5 Ev at ISO 200 circular fisheye lens: Sigma EX 4.5 mm) in this tray and with the lens pointing to the sky. The fluon-coated plate prevented the ants from crawling on the camera and lens. We obtained a horizontal position of the lens with a bubble level. For each nest, we took a hemispherical image with this setup and during overcast days. The photos were analyzed with the Hemiphot.R script to estimate canopy openness as the fraction of the sky unobscured by vegetation[55].

Prior to further analyses, we removed the variables nest activity and nest surface as they were strongly positively correlated with nest age (point-biserial correlation coefficient $r > 0.6$[56]).

**Joint species distribution model**
We analyzed the responses of the different red wood ant-associated functional groups simultaneously using joint species distribution modeling.

We built hierarchical joint species distribution models based on the Hierarchical Modeling of Species Communities (HMSC) framework described in ref. 57. HMSC is a multivariate Bayesian generalized linear mixed-effect model framework, which allows the simultaneous modeling of the responses of multiple species or functional groups and fits the residual correlation between these species or groups. We also considered the spatial structure of the communities (see Fig. 5) through spatially structured latent factors and accounted for the type of ant association (obligate vs facultative) as a trait. HMSC models with a lognormal link function were used to analyze the drivers of abundances of functional groups of myrmecophiles. Our analysis was restricted to the functional groups that were present in at least 30 of the 51 nests (seven obligate groups, and five facultative groups) to allow model convergence. As fixed effects, we included the predictors described above. Moisture, pH, and canopy openness were modeled as linear or second-order polynomial terms, and connectivity as a linear term only. We assumed the default prior distributions for all models. The posterior distribution was sampled using four Markov Chain Monte Carlo chains, each collecting 1000 samples, yielding a total of 4000 samples. We used a thinning rate of 1000 to avoid autocorrelation within the chains and excluded the first 500,000 iterations as burn-in, only sampling the subsequent 1,000,000 iterations per chain.

We evaluated Markov Chain Monte Carlo convergence by examining the distribution of the potential scale reduction factor of the parameters (equivalent to the Gelman-Rubin statistic). We compared 8 different HMSC models with all combinations of moisture, pH, and canopy openness modeled either as linear or second-order terms to determine the most optimal combination. The most optimal model was retained using the Widely Applicable Information Criterion (WAIC)[58]. The $\beta$ estimates of our model give the response of the functional groups to the predictors ("functional group niches"), and the $\gamma$ parameters model how functional group traits (here obligate vs facultative) influence their niches[57]. Explanatory power (how well the model explains the variation in the data) is assessed using $R^2$ when the models are fitted to all the data. To mitigate the risk of overfitting, we assessed the predictive power (how well the model can make accurate predictions on new data) of the retained model by conducting ten-fold cross-validation. The dataset was divided randomly into ten folds, each containing 5–6 nests. For each fold, the joint species distribution model was trained using 90% of the data from the other folds (designated as the training set), while the data from that specific fold served as the test data. The predictive power of the model was then evaluated by computing the average performance, measured as $R^2$, across the cross-validation folds. We evaluated the relative importance of the explanatory variables using variance partitioning analysis. Therefore, we grouped the explanatory variables into four categories: nest age, connectivity, environmental variables (moisture, pH, and canopy openness), and random spatial effect. The random spatial effect represents variation in species abundances which cannot be attributed to the variables included, and represents unmeasured environmental variation, random spatial effects as well as species co-occurrences.

**Community differences and evenness**
We analyzed whether nest age influenced the community composition. First, we plotted with an ordination the community differences using an NMDS (BC dissimilarities functional group abundances, vegan package[59]). Then, we ran a permutational multivariate analysis of variance (PERMANOVA) with the function adonis2 of the vegan package with 999 permutations[59]. The response matrix contained the pairwise BC dissimilarities between the nests based on the functional group abundances. Nest age, connectivity, and the interaction between connectivity and nest age were included as predictors. Next, we calculated the Pielou evenness index $J$ for each nest. This index measures how homogeneously or evenly the individuals are distributed over the different functional groups in the symbiont community. $J$ is bounded between 0 (no evenness) and 1 (complete evenness). We fitted ($J$) against the predictor's nest age, connectivity, and interaction using a betaregression model, which is the most appropriate model for proportional data. Residuals were examined with diagnostic plots

to confirm model fit and no violations were detected. We also did not record variable dispersion or spatial autocorrelation in this model. The significance of the predictors was tested with a Wald test. Finally, we visually compared the rank abundance curves of old and new nests and plotted the averaged symbiont food webs of old and new nests using the package igraph[60]. Trophic interactions of obligate groups were based on refs. 41,52, the diet of facultative groups was derived from literature[53].

## Stability analysis

We started from a general adjacency matrix $A$, representing overall food web relationships (see Supplementary Table S2). Rows were resources and columns were consumers: when an element $a_{ij} = 1$ signifies that $i$ consumed $j$. This matrix included spider$_H$, rove beetle$_P$, rove beetle$_{SP1}$, rove beetle$_{SP2}$, rove beetle$_S$, beetle$_D$, beetle$_B$, springtail$_D$, isopod$_D$, fac armoured$_D$, fac pred$_{P1}$, fac pred$_{P2}$, fac prey$_D$, fac rove beetle$_S$, fac mite$_D$, fac mite$_P$, ant brood, and nest food. Ant brood and nest food are two aggregate basal resources, representing a mix of various specific resources (ant brood: eggs, larvae, pupae, nest food: dead ant corpses, organic nest material, fungi, and spores). We assumed that competition between two consumers happened when they both fed on the same aggregate resource. For every nest, we created a local adjacency food web matrix by subsetting the general adjacency matrix. These local adjacency matrices included only the functional groups present at that nest. We also excluded a functional group from the local adjacency matrix if it was rare in the nest, i.e., when its abundance fell below a threshold equivalent to the 0.1 quantile of the observed nest abundances for that functional group. For every local adjacency matrix, we then generated 50 random matrices, where we sampled $a_{ij} \sim N(\mu, \sigma)$ with $\mu = -0.1$ if it represented the effect of a consumer on its resource, or the strength of competition for basal resources between two taxa. We took $\mu = 0.1$ when it represented the effect of a resource on its consumer. In case a taxon preyed on a competitor, the generated effects of consumption and competition were summed, $\sigma$ was always 0.05. Setting $\mu$ and $\sigma$ to other values did not change the results (Supplementary Fig. S4). Finally, all diagonal elements were set to $-1$. Stability was computed as the local asymptotic stability, a method that assesses whether and how fast a system when perturbed infinitesimally away from an equilibrium will eventually return to its original state[2]. We assumed, following[61], that the local dynamics in the nest operate at a much faster rate than invasions, which allows the community to have sufficient time to reach its configuration before the next invasion arrives. The stability analysis relies on examining the largest real-valued eigenvalue of the community matrix, denoted as $\Re(\lambda_1)$. In cases where the largest real-valued eigenvalue is positive, it indicates an unstable system. Conversely, if the largest real eigenvalue is negative, it signifies that the system will eventually return to equilibrium following perturbation with more negative values indicating a quicker return to stability for the system. Local asymptotic stability as assessed by $\Re(\lambda_1)$ is hereafter called stability. In the case of Lotka-Volterra dynamics, the community matrix $M$ is simply $DA$, where $D$ is a diagonal matrix with the equilibrium relative abundances as its entries. The use of relative abundances was needed because older nests supported a higher total number of individuals, inherently leading to higher eigenvalues of the community matrix (if a matrix is multiplied by a scalar, then all its eigenvalues are multiplied by the same scalar).

Nest stability was assessed by computing the average of the 50 $\Re(\lambda_1)$ values linked to the 50 permuted community matrices calculated for each nest. May's stability criterion for random matrices indicates that higher species diversity has a destabilizing effect on communities[2,45]. To be able to compare the stability of our nest communities with different sizes, we need to control for ecosystem size (number of functional groups in the community). Therefore, we ran a general linear model not only with the predictor nest age (two levels: old vs. new) but also with the continuous predictor community size (number of functional groups present) and the interaction between nest age and community size. The logarithm of the absolute value of the nest-averaged $\Re(\lambda_1)$ was modeled as the dependent variable. We did not include nests with a lower community size than six because these were only present in the new nest category. We then wanted to

examine to what extent putative relationships between nest age and stability were driven by the trophic network structure: i.e., the distribution of relative abundances ($D$) or by the specific topology (arrangement of the interactions, who eats whom) of the interaction matrix ($A$). To this end, we created two versions of $D$: one containing the observed relative abundances, and one with perfectly even abundances (i.e., diagonal element $d_i = 1/n \, \forall \, i$). We also created two versions for $A$: one containing the observed interactions (observed topology), and one representing a randomly rewired version (randomized topology). Random rewiring was done by shuffling all off-diagonal elements. The analysis for the scenario with observed relative abundances and observed topological arrangement of the nodes is explained above. For the three other scenarios, i.e., observed relative abundances/randomized topology, even abundances/observed topology, and even abundances/randomized topology, we also ran a similar general linear model with lambda as response, and nest age * community size as predictors. Residuals and assumptions of the four models were checked using the DHARMa package[62]. Data and code are available via figshare[63].

## Data availability

All data are available via figshare: https://doi.org/10.6084/m9.figshare.24799911.

## Code availability

All codes are available via figshare: https://doi.org/10.6084/m9.figshare.24799911.

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

## Acknowledgements

We thank Pieter Vantieghem for his help with sorting the arthropods from the pitfalls and Diederik Strubbe for kindly giving access to his high-performance computer. We are indebted to the Agency of Nature and Forest (ANB) for sampling permission at the study site. This work was supported by Fonds Wetenschappelijk Onderzoek Vlaanderen—FWO (1203020N to T.P.) and Fonds de la Recherche Scientifique—FNRS (Chargé de recherches 30257865 to T.P.). We thank two anonymous reviewers for their helpful and constructive comments.

## Author contributions

T.P., D.B., F.D.L. designed the study, T.P. collected field data, and performed field data analysis. F.D.L. carried out the stability analysis. T.P. and F.D.L. wrote the manuscript and all authors contributed substantially to revisions.

## Competing interests

The authors declare no competing interests.
