## [Peer Review File · Communications Biology]

Reviewers' comments:

Reviewer #1 (Remarks to the Author):

This study focuses on whether and how alterations of network structure affect stability during succession in complex communities. Using the striking symbiotic arthropod communities forming food webs inside red wood ant nests, it shows a significant reorganization of the symbiont community between young and old nests, with greater homogeneity of functional groups playing a crucial role in enhancing the dynamic stability of the symbiont food web associated with old nests.

Although I must admit that I am not competent to evaluate all the models used in these analyses (which seem, in most cases, far too technical for me), such a detailed study is commendable and gives interesting insights into the differences in factors affecting community composition between old and new nests. The general methodology for determining the predictors (nest area, activity, nest age, connectivity, humidity, pH and canopy openness) was sophisticated and represented a huge amount of work. However, some of the data analysis procedures require further explanation. For example, the exclusion of individuals from rare species or groups from the analyses (see lines 80-82, 297-298, 304-305) should be justified. Similarly, it is not clear why *Quedius brevis* and *Thiasophila angulata* were considered as two different obligate functional groups (rove beetleSP1 and rove beetleSP2), whereas *Amidobia talpa*, *Dinarda maekelii*, *Notothecta flavipes* and *Lyprocorrhe anceps* or *Monotoma angusticollis* and *Monotoma conicicollis* were grouped together in a single obligate functional group (rove beetleS and beetleD, respectively) (lines 290-297). Such apparently artificial differentiation could influence the homogeneity of functional groups.

Somme other minor comments:

- line 22: "affect" not "affects"

- line 43: "linked to their specific" not "linked to to their specific"

- lines 71 and 174: As indicated in lines 103-105, treating nest age as a continuous predictor was not totally conclusive with your model. Rather than saying "as the host nest ages" or "as nests age", which give the idea of a dynamic phenomenon unsupported by your model, it would be more correct to say "between young and old nests".

- line 86: "diagnostics" not "diagnostistics"

- line 162: "stability of old and new nests was also" not "stability of old and new nests were also".

- lines 161-163: The meaning of this sentence is confusing. How can the stability of old and new nests overlap?

- lines 176-177 and 229-230: Do you have any ideas or hypotheses to explain why larger symbiont communities don't follow this pattern of greater stability when the nests are older and why stability in old nest communities decreased faster with increasing community size than in new nest communities?

- line 197: “in older nests” not “with successional age”. See previous comment on lines 71 and 174.
- line 239: “well-established” not “well established”.
- line 298: “were not included” not “were not not included”
- line 369: “explains” not “explain”
- line 470: “All data and code are available” not “All data and code is available”
- line 478: “Ecological networks” not “Review: ecological networks”.
- line 486: “Brännström, Å” not “Brännström, A”.
- line 500: “Ecography 2022, e06112 (2022)” not “Ecography 2022 (2022)”.
- line 524: “6, 881 (2015)” not “6, 1–14 (2015)”.
- line 594: Reference [41] is lacking.
- line 610: Delete “: The Journal of the Society for Conservation Biology”.
- line 614: “Robinson, E. J. H., Stockan, J. A. & Iason, G. R. Wood ants and their interaction with other organisms, pp. 177–206. In Stockan J. A. & Robinson E. J. H. (eds) Wood ant ecology and conservation. Cambridge University Press, Cambridge (2016).” not “Robinson, E. J., Stockan, J. A. & Iason, G. R. 8 r wood ants and their interaction with other organisms Wood ant ecology and conservation 177 (2016).”.
- line 630: “thesis, KU Leuven” not “thesis, Kuleuven”.
- line 643: “Csárdi” not “Csardi”.

Reviewer #2 (Remarks to the Author):

In the manuscript “A successional shift enhances stability in ant symbiont communities”, the authors evaluated the effect of nest age, and the environment on communities of symbionts of ant nests. They also created food webs for these communities, and evaluated the stability of these local food webs. They found that community composition is largely driven by the age of the nest, where older nests contain more facultative symbionts. Further, they found that older communities were more stable.

Overall, I really liked this paper. The dataset is very unique, and these communities of symbionts are an excellent system to evaluate food web assembly mechanisms.

Major comments:

- The analyses for the effect of the environment on community composition are well done, and are very standard. It would be nice to have a multivariate analysis plot showing community composition differences between old and new nests, even if this is in the supplementary material.

- Regarding the creation of the food webs, it was unclear to me what data or resources were used to create the general adjacency matrix. Was it feeding experiments, gut content analysis, the literature, etc. This part needs more clarity. It would also be helpful if the authors show what the general adjacency matrix looks like in the supplementary material.

- I am not entirely convinced with the local asymptotic stability analysis, mostly because this type of analysis depends heavily on the interaction strengths, and while the adjacency matrix were constructed with knowledge about the system, the interaction strengths themselves were very arbitrary. The authors state: "For every local adjacency matrix, we then generated 50 random matrices, where we sampled $a_{ij} \sim N(\mu, \sigma)$ with $\mu = -0.1$ if it represented the effect of a consumer on its resource, or the strength of competition for basal resources between two taxa. We took $\mu = 0.1$ when it represented the effect of a resource on its consumer. In case a taxon preyed on a competitor, the generated effects of consumption and competition were summed, σ was always 0.05. Setting μ and σ to other values did not change the results. Finally, all diagonal elements were set to -1.". The authors need to demonstrate that the choice of $\mu = 0.1$ is not biasing the results. Having a sensitivity analysis here would go a long way at convincing the reader that the results are robust.

- Further, from the methods it is somewhat unclear how the relative abundance is incorporated into the construction of the adjacency matrix, and the local asymptotic stability analysis. The manuscript could use more detail here.

- The discussion should also go over the results presented for figures 1-3 and not be so entirely focused on the stability analysis.

Minor comments:

- Is community size in Figure 4 simply the number of species in the food web? This is not entirely clear. Presumably most of the additional species added are facultative? The authors could explain this better.

Dear reviewers,

Detailed point-by-point responses to your comments can be found below. Changes in the manuscript are highlighted in the revised manuscript in blue.

Sincerely,

Dr. Thomas Parmentier on behalf of all co-authors

Dear Dr Parmentier,

Your manuscript entitled "A successional shift enhances stability in ant symbiont communities" has now been seen by 2 referees. You will see from their comments below that while they find your work of considerable interest, some important points are raised. We are interested in the possibility of publishing your study in Communications Biology, but would like to consider your response to these concerns in the form of a revised manuscript before we make a final decision on publication.

We therefore invite you to revise and resubmit your manuscript, taking into account the points raised.

Reviewer #1 (Remarks to the Author):

This study focuses on whether and how alterations of network structure affect stability during succession in complex communities. Using the striking symbiotic arthropod communities forming food webs inside red wood ant nests, it shows a significant reorganization of the symbiont community between young and old nests, with greater homogeneity of functional groups playing a crucial role in enhancing the dynamic stability of the symbiont food web associated with old nests.

Although I must admit that I am not competent to evaluate all the models used in these analyses (which seem, in most cases, far too technical for me), such a detailed study is commendable and gives interesting insights into the differences in factors affecting community composition between old and new nests. The general methodology for determining the predictors (nest area, activity, nest age, connectivity, humidity, pH and canopy openness) was sophisticated and represented a huge amount of work.

Thank you for your appreciation and for acknowledging the amount of work it entailed.

However, some of the data analysis procedures require further explanation.

For example, the exclusion of individuals from rare species or groups from the analyses (see lines 80-82, 297-298, 304-305) should be justified.

Lines 80-81: As the Results sections comes before the Method section in Communications Biology, it is confusing to mention here that individuals were excluded. Therefore we only list the total number of individuals collected at the beginning of the results section. We believe it's crucial to emphasize these numbers to underscore the comprehensiveness of this study. Details on the very small fraction of individuals that were not included in the analyses can be found under the Method section. See directly below:

Lines 301-303: These obligate myrmecophiles were rare (exact numbers can be found via figshare: raw data) and should all be assigned to different functional groups based on their taxonomy and diet (cf Parmentier et al. 2016, Oikos). This would result in an excessive number of functional groups to manage effectively, caused convergence issues with the JSDM and made the stability analysis much more complex.

We added here: *"The rare obligate species Myrmetes paykullii (N = 4), Pella humeralis (N = 7), Emphylyus glaber (N = 5), and Leptacinus formicetorum (N = 2) excluded from our analyses to limit the number of obligate functional groups. By doing so, we avoided convergence issues in our analyses and reduced the complexity of the theoretical stability analyses (see below)."*

Lines 310-313: The same reasoning applies in this case. Including these organisms would increase the number of functional groups and complicate the analyses. Moreover it is unknown what type of food nest-associated Hymenoptera and Lepidoptera feed on. We added here: *"Rare facultative species (<15 individuals over all nests, e.g. Heteroptera, Hymenoptera, Lepidoptera, a total of 86 individuals) were not included to limit the number of facultative functional groups. In addition, the trophic interactions of these groups were mostly unclear. The trophic ecology and trophic interactions of the different functional groups are given in Table S2"*

Similarly, it is not clear why *Quedius brevis* and *Thiasophila angulata* were considered as two different obligate functional groups (rove beetleSP1 and rove beetleSP2), whereas *Amidobia talpa*, *Dinarda maekelii*, *Notothecta flavipes* and *Lyprocorrhe anceps* or *Monotoma angusticollis* and *Monotoma conicollis* were grouped together in a single obligate functional group (rove beetleS and beetleD, respectively) (lines 290-297). Such apparently artificial differentiation could influence the homogeneity of functional groups.

We want to stress that the differentiation was well-thought and not artificial. The differentiation in 16 functional groups was based on the following three criteria:

- ➔ Obligate or facultative association
- ➔ Trophic interactions
- ➔ Taxonomic group

We now included the adjacency matrix (trophic matrix) directly in the electronic supplementary material (Table S2) and also refer to this when presenting the functional groups. We also made the text clearer and explain better the rationale behind the choice of functional groups. We stress that related species (same taxonomic group) with a different trophic ecology are placed into different functional groups. For example both rove beetle$_{SP1}$ (*Quedius brevis*) and rove beetle$_{SP2}$ (*Thiasophila angulata*) feed/hunt for different species, but rove beetle$_{SP1}$ does not feed on ant brood in contrast to rove beetle$_{SP2}$ (*Thiasophila angulata*). Rove beetle$_{S}$ encompass different rove beetle species that scavenge, but none of these species were observed to feed on living individuals (<-> rove beetle$_{SP1}$ and rove beetle$_{SP2}$). Facultative rove beetles (fac rove beetles$_{S}$) have similar trophic interactions than some obligate rove beetles (rove beetle$_{S}$) but were assigned to a different functional group to be able to discriminate obligate and facultative groups (Table S2). *Monotoma* beetles (beetle$_{D}$) belong to a different beetle family than rove beetles and also have different trophic interactions (see Table S2). That is why it seems justified to categorize them into a distinct functional group.
We added L297-300: "*Species that are taxonomically related but exhibit distinct trophic ecologies are categorized into separate functional groups: for example, rove beetles are assigned to rove beetle$_{P}$, rove beetles$_{S}$, rove beetle$_{SP1}$ or rove beetle$_{SP2}$ based on their specific diet (functional group-specific diet in Table S2).*"

*Table S2. General adjacency matrix A, representing overall food web relationships between 16 functional symbiont groups. Rows are resources and columns are consumers. An element aij = 1 signifies that i consumed j, when aij = 0 no trophic interaction is assumed between i and j.*

| | spider$_{P}$ | rove beetle$_{P}$ | rove beetle$_{SP1}$ | rove beetle$_{SP2}$ | rove beetles$_{S}$ | beetle$_{D}$ | beetle$_{B}$ | springtail$_{D}$ | isopod$_{D}$ | fac armoured$_{D}$ | fac pred$_{P1}$ | fac pred$_{P2}$ | fac prey$_{D}$ | fac rove beetles$_{S}$ | fac mite$_{D}$ | fac mite$_{P}$ | ant brood | nest food |
|---|---|---|---|---|---|---|---|---|---|---|---|---|---|---|---|---|---|---|
| spider$_{P}$ | NaN | 0 | 1 | 0 | 0 | 0 | 0 | 0 | 0 | 0 | 1 | 0 | 0 | 0 | 0 | 0 | 0 | 0 |
| rove beetle$_{P}$ | 1 | NaN | 1 | 1 | 0 | 0 | 0 | 0 | 0 | 0 | 1 | 0 | 0 | 0 | 0 | 0 | 0 | 0 |
| rove beetle$_{SP1}$ | 0 | 0 | NaN | 0 | 0 | 0 | 0 | 0 | 0 | 0 | 0 | 0 | 0 | 0 | 0 | 0 | 0 | 0 |
| rove beetle$_{SP2}$ | 1 | 0 | 1 | NaN | 0 | 0 | 0 | 0 | 0 | 0 | 1 | 0 | 0 | 0 | 0 | 0 | 0 | 0 |
| rove beetles$_{S}$ | 1 | 0 | 1 | 1 | NaN | 0 | 0 | 0 | 0 | 0 | 1 | 0 | 0 | 0 | 0 | 0 | 0 | 0 |
| beetle$_{D}$ | 1 | 0 | 1 | 1 | 0 | NaN | 0 | 0 | 0 | 0 | 1 | 0 | 0 | 0 | 0 | 0 | 0 | 0 |
| beetle$_{B}$ | 0 | 0 | 0 | 0 | 0 | 0 | NaN | 0 | 0 | 0 | 0 | 0 | 0 | 0 | 0 | 0 | 0 | 0 |
| springtail$_{D}$ | 1 | 1 | 0 | 0 | 0 | 0 | 0 | NaN | 0 | 0 | 1 | 1 | 0 | 0 | 0 | 1 | 0 | 0 |
| isopod$_{D}$ | 0 | 0 | 1 | 0 | 0 | 0 | 0 | 0 | NaN | 0 | 1 | 0 | 0 | 0 | 0 | 0 | 0 | 0 |
| fac armoured$_{D}$ | 0 | 0 | 0 | 0 | 0 | 0 | 0 | 0 | 0 | NaN | 0 | 0 | 0 | 0 | 0 | 0 | 0 | 0 |
| fac pred$_{P1}$ | 0 | 0 | 0 | 0 | 0 | 0 | 0 | 0 | 0 | 0 | NaN | 0 | 0 | 0 | 0 | 0 | 0 | 0 |
| fac pred$_{P2}$ | 1 | 1 | 1 | 0 | 0 | 0 | 0 | 0 | 0 | 0 | 1 | NaN | 0 | 0 | 0 | 0 | 0 | 0 |
| fac prey$_{D}$ | 1 | 1 | 1 | 1 | 0 | 0 | 0 | 0 | 0 | 0 | 1 | 1 | NaN | 0 | 0 | 1 | 0 | 0 |
| fac rove beetle$_{S}$ | 1 | 0 | 1 | 1 | 0 | 0 | 0 | 0 | 0 | 0 | 1 | 0 | 0 | NaN | 0 | 0 | 0 | 0 |
| fac mite$_{D}$ | 1 | 1 | 1 | 1 | 0 | 0 | 0 | 0 | 0 | 0 | 1 | 1 | 0 | 1 | NaN | 1 | 0 | 0 |
| fac mite$_{P}$ | 1 | 1 | 1 | 1 | 0 | 0 | 0 | 0 | 0 | 0 | 1 | 1 | 0 | 1 | 0 | NaN | 0 | 0 |
| ant brood | 0 | 0 | 0 | 1 | 1 | 1 | 1 | 0 | 1 | 0 | 0 | 0 | 0 | 0 | 0 | 0 | 0 | 0 |

nest food	0	0	1	1	1	1	1	1	1	1	1	1	1	1	1	1	0	0
-----------	---	---	---	---	---	---	---	---	---	---	---	---	---	---	---	---	---	---

Somme other minor comments:

- line 22: “affect” not “affects”

This typo was corrected (L 22)

- line 43: “linked to their specific” not “linked to to their specific”

This typo was corrected (L 43)

- lines 71 and 174: As indicated in lines 103-105, treating nest age as a continuous predictor was not totally conclusive with your model. Rather than saying "as the host nest ages" or "as nests age", which give the idea of a dynamic phenomenon unsupported by your model, it would be more correct to say "between young and old nests".

Indeed, this is a valid comment. We replaced as nests ages by “between new and old nests” (L 71, L174)

- line 86: “diagnostics” not “diagnostistics”

This typo was corrected (L 85)

- line 162: “stability of old and new nests was also” not “stability of old and new nests were also”.

In the rephrased sentence, “were” is correct (L 162), see following comment.

- lines 161-163: The meaning of this sentence is confusing. How can the stability of old and new nests overlap?

Indeed, a good point. We mean that the stability values of old and new nests overlap. We rephrased this in the manuscript (L 162)

- lines 176-177 and 229-230: Do you have any ideas or hypotheses to explain why larger symbiont communities don't follow this pattern of greater stability when the nests are older and why stability in old nest communities decreased faster with increasing community size than in new nest communities?

This is an interesting question: one possible explanation is that small communities are stronger connected in new nests than old nests and have larger variation in interaction strength, whereas connectance and variation in interaction strength become comparable in old and new large communities. These are two factors that are known to destabilize networks (see May 1972).

- line 197: “in older nests” not “with successional age”. See previous comment on lines 71 and 174.

Indeed, this is a valid comment. We replaced “with successional age” by “in older nests” (L 197)

- line 239: “well-established” not “well established”.

This typo was corrected (L 239)

- line 298: “were not included” not “were not not included”

This typo was corrected (L 301)

- line 369: “explains” not “explain”

This typo was corrected (L 378)

- line 470: “All data and code are available” not “All data and code is available”

This typo was corrected (L 489)

- line 478: “Ecological networks” not “Review: ecological networks”.

This typo was corrected (L 498)

- line 486: “Brännström, Å” not “Brännström, A”.

This typo was corrected (L 506)

- line 500: “Ecography 2022, e06112 (2022)” not “Ecography 2022 (2022)”.

This info was corrected (L 520)

- line 524: “6, 881 (2015)” not “6, 1–14 (2015)”.

This info was corrected (L 544)

- line 594: Reference [41] is lacking.

We added this reference (L 614), this reference was listed as reference [52]. Therefore, references with numbers higher than 52 changed.

- line 610: Delete “: The Journal of the Society for Conservation Biology”.

We deleted this part (L 627)

- line 614: “Robinson, E. J. H., Stockan, J. A. & Iason, G. R. Wood ants and their interaction with other organisms, pp. 177–206. In Stockan J. A. & Robinson E. J. H. (eds) Wood ant ecology and conservation. Cambridge University Press, Cambridge (2016).” not “Robinson, E. J., Stockan, J. A. & Iason, G. R. 8 r wood ants and their interaction with other organisms Wood ant ecology and conservation 177 (2016).”.

We corrected this reference (L 636)

- line 630: “thesis, KU Leuven” not “thesis, Kuleuven”.

This typo was corrected (L 648)

- line 643: “Csárdi” not “Csardi”.

This typo was corrected (L 662)

Reviewer #2 (Remarks to the Author):

In the manuscript “A successional shift enhances stability in ant symbiont communities”, the authors evaluated the effect of nest age, and the environment on communities of symbionts of ant nests. They also created food webs for these communities, and evaluated the stability of these local food webs. They found that community composition is largely driven by the age of the nest, where older nests contain more facultative symbionts. Further, they found that older communities were more stable.

Overall, I really liked this paper. The dataset is very unique, and these communities of symbionts are an excellent system to evaluate food web assembly mechanisms.

Thank you for your appreciation!

Major comments:

- The analyses for the effect of the environment on community composition are well done, and are very standard. It would be nice to have a multivariate analysis plot showing community composition differences between old and new nests, even if this is in the supplementary material.

We conducted a multivariate analysis (NMDS) and added a plot in the suppl material (Fig S2) This analysis further supports our main conclusions. Communities of old and new nests are different. In old nests, obligate functional groups become relatively more important in the community.

Figure S2. Multivariate analysis of the 51 symbiont communities (from 51 nests) using Non-metric dimensional scaling. Only the predictor nest age was significant. Communities in old nests are given in orange dots, those of new nests in gray dots. Number codes that are underlined refer to obligate functional groups found in the communities: 4) springtail_D, 6) spider_P, 8) beetle_B, 9) rove beetle_P, 10) beetle_D, 11) rove beetle_{SP2}, 12) rove beetle_S, 13) isopod_D, 16) rove beetle_{SP1}. Number codes that are not underlined represent facultative functional groups found in the communities: 1) fac mite_D, 2) fac mite_P, 3) fac prey_D, 5) fac armoured_B, 7) fac rove beetle_S, 14) fac pred_{P1}, 15) fac pred_{P2}. Number codes of the functional groups are similar to those used in Fig. 3.

- Regarding the creation of the food webs, it was unclear to me what data or resources were

used to create the general adjacency matrix. Was it feeding experiments, gut content analysis, the literature, etc. This part needs more clarity. It would also be helpful if the authors show what the general adjacency matrix looks like in the supplementary material.

This indeed needs more clarification that we now added in L409: “Trophic interactions of obligate groups were based on^{41,53}, diet of facultative groups was derived from literature⁶⁰.”

We also included the general adjacency matrix (Table S2) in the supplementary material, see also comment of Reviewer 1.

- I am not entirely convinced with the local asymptotic stability analysis, mostly because this type of analysis depends heavily on the interaction strengths, and while the adjacency matrix were constructed with knowledge about the system, the interaction strengths themselves were very arbitrary. The authors state: “For every local adjacency matrix, we then generated 50 random matrices, where we sampled $a_{ij} \sim N(\mu, \sigma)$ with $\mu = -0.1$ if it represented the effect of a consumer on its resource, or the strength of competition for basal resources between two taxa. We took $\mu = 0.1$ when it represented the effect of a resource on its consumer. In case a taxon preyed on a competitor, the generated effects of consumption and competition were summed, σ was always 0.05. Setting μ and σ to other values did not change the results. Finally, all diagonal elements were set to -1.” The authors need to demonstrate that the choice of $\mu = 0.1$ is not biasing the results. Having a sensitivity analysis here would go a long way at convincing the reader that the results are robust.

We agree that this needed more info. Therefore we now included a sensitivity analysis where we reran our analysis with different μ -values. The results are presented in Fig. S4 and referred to in the text (L164, 430). Lower or higher values of μ did not affect the general conclusions and outline of results presented in Fig. 4. Also the statistical differences hold (Fig. S4). In theory you can take very high μ 's approaching -1, but we don't think this is realistic for this multicomunity system, where functional groups do not have one preferred food resource. Importantly, we set the level of self-limitation (diagonal) higher than μ , otherwise the community would be inherently unstable.

Figure S4. Prediction of local stability for symbiont food webs in old and new nests with increasing community size (= number of functional groups) in four scenarios and with different μ . Interaction coefficients were drawn from $a_{ij} \sim N(\mu, \sigma)$ with $\mu = -0.10$ and $\sigma = 0.05$ in the main manuscript (Fig. 4). Here we conducted a sensitivity analysis with $a_{ij} \sim N(\mu, \sigma)$ and set μ to the values : a) $\mu = -0.05$, b) $\mu = -0.15$, c) $\mu = -0.20$ and d) $\mu = -0.25$. For the scenario with observed relative abundances and observed topology, older nests exhibited significantly greater stability compared to new nests, regardless of the parameter μ (corresponding P-values displayed on plot).

- Further, from the methods it is somewhat unclear how the relative abundance is incorporated into the construction of the adjacency matrix, and the local asymptotic stability analysis. The manuscript could use more detail here.

In the first version, we explained this by:

“The use of relative abundances was needed because eigenvalues scale with the magnitude of a matrix’ entries, and because older nests had more individuals.”

We rephrased this a bit and added:

L442-445: *The use of relative abundances was needed because older nests supported a higher total number of individuals in total, leading to higher eigenvalues of the community matrix (if a matrix is multiplied by a scalar, then all its eigenvalues are multiplied by the same scalar).*

The effect of a scalar on eigenvalues (~ stability) can be easily shown by a simple example:

Given: A random matrix A with interactions between three species

$$A = \begin{pmatrix} 1.00 & -0.20 & -0.15 \\ -0.08 & -1.00 & -0.16 \\ -0.08 & 0.01 & -1.00 \end{pmatrix}$$

Let's say the densities of the three species are 3, 6, and 1, represented by a diagonal matrix $D1$

$$D1 = \begin{pmatrix} 3 & 0 & 0 \\ 0 & 6 & 0 \\ 0 & 0 & 1 \end{pmatrix}$$

To compute the community matrix $M1$, we need to multiply $D1$ with A . Lambda max of this community matrix $\lambda_1 = -0.988$

Let's say we multiply the abundances of the community by a factor of 10 (three species: 30, 60 and 10, given as diagonal matrix $D2$)

$$D2 = \begin{pmatrix} 30 & 0 & 0 \\ 0 & 60 & 0 \\ 0 & 0 & 10 \end{pmatrix}$$

To compute the community matrix $M2$, we multiply $D2$ with A . Lambda max of this community matrix $\lambda_2 = -9.880$. $\rightarrow \lambda_1 * 10$

This simple example shows that eigenvalues scale with the absolute densities of the species in the community.

- The discussion should also go over the results presented for figures 1-3 and not be so entirely focused on the stability analysis.

We respectfully disagree with the referee: The discussion contains 5 paragraphs:

1st: summarizing the main results, (L173-177)

2nd: change in composition in old nests (L178-192)

3rd: higher evenness + specialization in old nests (L193-212)

4th: effects of community change on stability (L213-231)

5th: concluding remarks (L233-240)

Paragraph 2 and 3 cover the results presented in figures 1-3 and do not focus on the stability analysis. This is only covered in paragraph 4. We now added direct references to the figures in the discussion to make this clearer (L192, 197, 202).

Minor comments:

- Is community size in Figure 4 simply the number of species in the food web? This is not entirely clear. Presumably most of the additional species added are facultative? The authors could explain this better.

Community size is the number of functional groups (see Fig.3: 16 functional groups in total) in the food web. We try to explain this better throughout the manuscript. L 150, 167, 228, 450, 452

REVIEWERS' COMMENTS:

Reviewer #1 (Remarks to the Author):

The information provided by the authors regarding my previous comments is perfectly clear and answers the few questions I raised. The few lines added to the text provide the clarifications needed for a proper understanding of the manuscript. Excellent work.

Just a doubt about line 310: is it "<15 individuals" or "<15 species"?

Jean-Paul Lachaud

Reviewer #2 (Remarks to the Author):

Thank you for your review, you have addressed all of my comments sufficiently. I really like the new NMDS plot and the sensitivity analysis.

Dear referee,

We clarified the last minor unclarity:

Reviewer#1

Just a doubt about line 310: is it "<15 individuals" or "<15 species"?

It is 15 individuals as was indicated in the original text. We rephrased the sentence to clarify and avoid misunderstandings.

L284: Rare facultative groups (= groups with <15 individuals over all nests, e.g. Heteroptera, Hymenoptera, Lepidoptera, these groups compromised a total of 86 individuals) were not included to limit the number of facultative functional groups.